# A Multi-Factorial Observational Study on Sequential Fecal Microbiota Transplant in Patients with Medically Refractory *Clostridioides difficile* Infection

**DOI:** 10.3390/cells10113234

**Published:** 2021-11-19

**Authors:** Tanya M. Monaghan, Niharika A. Duggal, Elisa Rosati, Ruth Griffin, Jamie Hughes, Brandi Roach, David Y. Yang, Christopher Wang, Karen Wong, Lynora Saxinger, Maja Pučić-Baković, Frano Vučković, Filip Klicek, Gordan Lauc, Paddy Tighe, Benjamin H. Mullish, Jesus Miguens Blanco, Julie A. K. McDonald, Julian R. Marchesi, Ning Xue, Tania Dottorini, Animesh Acharjee, Andre Franke, Yingrui Li, Gane Ka-Shu Wong, Christos Polytarchou, Tung On Yau, Niki Christodoulou, Maria Hatziapostolou, Minkun Wang, Lindsey A. Russell, Dina H. Kao

**Affiliations:** 1NIHR Nottingham Biomedical Research Centre, University of Nottingham, Nottingham NG7 2UH, UK; 2Nottingham Digestive Diseases Centre, School of Medicine, University of Nottingham, Nottingham NG7 2UH, UK; ruth.griffin1@nottingham.ac.uk; 3MRC-Arthritis Research UK Centre for Musculoskeletal Ageing Research, Institute of Inflammation and Ageing, University of Birmingham, Birmingham B15 2TT, UK; n.arora@bham.ac.uk; 4Institute of Clinical Molecular Biology, Universitätsklinikum Schleswig-Holstein, Christian-Albrecht University of Kiel, 24105 Kiel, Germany; e.rosati@ikmb.uni-kiel.de (E.R.); a.franke@mucosa.de (A.F.); 5Synthetic Biology Research Centre, The University of Nottingham Biodiscovery Institute, University of Nottingham, Nottingham NG7 2RD, UK; svzjh2@exmail.nottingham.ac.uk; 6Division of Gastroenterology, Department of Medicine, University of Alberta; Edmonton, Alberta, AB T6G 2G3, Canada; brandi.roach@albertahealthservices.ca (B.R.); dyyang@ualberta.ca (D.Y.Y.); cwang1@ualberta.ca (C.W.); kwong3@ualberta.ca (K.W.); 7Division of Infectious Diseases, Department of Medicine, University of Alberta; Edmonton, Alberta, AB T6G 2G3, Canada; saxinger@ualberta.ca; 8Glycoscience Research Laboratory, Genos Ltd., Borongajska cesta 83H, 10000 Zagreb, Croatia; mpucicbakovic@genos.hr (M.P.-B.); fvuckovic@genos.hr (F.V.); fklicek@genos.hr (F.K.); glauc@genos.hr (G.L.); 9Faculty of Pharmacy and Biochemistry, University of Zagreb, 10000 Zagreb, Croatia; 10School of Life Sciences, University of Nottingham, Nottingham NG7 2RD, UK; mrzpjt@exmail.nottingham.ac.uk; 11Division of Digestive Diseases, Department of Metabolism, Digestion and Reproduction, Faculty of Medicine, Imperial College London, London SW7 2AZ, UK; b.mullish@imperial.ac.uk (B.H.M.); j.miguens-blanco18@imperial.ac.uk (J.M.B.); julie.mcdonald@imperial.ac.uk (J.A.K.M.); j.marchesi@imperial.ac.uk (J.R.M.); 12MRC Centre for Molecular Bacteriology and Infection, Imperial College London, London SW7 2AZ, UK; 13School of Veterinary Medicine and Science, University of Nottingham, Nottingham NG7 2UH, UK; ning.xue@microlise.com (N.X.); svztd@exmail.nottingham.ac.uk (T.D.); 14College of Medical and Dental Sciences, Institute of Cancer and Genomic Sciences, Centre for Computational Biology, University of Birmingham, Birmingham B15 2TT, UK; a.acharjee@bham.ac.uk; 15Shenzhen Digital Life Institute, Shenzhen 518016, China; liyr@icarbonx.com; 16Department of Biological Sciences, Department of Medicine, University of Alberta, Edmonton, AB T6G 2E1, Canada; gane@ualberta.ca; 17Department of Biosciences, John van Geest Cancer Research Centre, Centre for Health Aging and Understanding Disease, School of Science and Technology, Nottingham Trent University, Nottingham NG11 8NS, UK; Christos.polytarchou@ntu.ac.uk (C.P.); payton.yau@ntu.ac.uk (T.O.Y.); niki.christodoulou2019@my.ntu.ac.uk (N.C.); maria.hatziapostolou@ntu.ac.uk (M.H.); 18Innovation Lab, Innovent Biologics, Inc., Suzhou 215011, China; 19Division of Gastroenterology, Department of Medicine, McMaster University, Hamilton, ON L8N 3Z5, Canada

**Keywords:** fecal microbiota transplantation, *Clostridioides difficile*, immunosenescence, host-microbial interactions, systems biology

## Abstract

Fecal microbiota transplantation (FMT) is highly effective in recurrent *Clostridioides difficile* infection (CDI); increasing evidence supports FMT in severe or fulminant *Clostridioides difficile* infection (SFCDI). However, the multifactorial mechanisms that underpin the efficacy of FMT are not fully understood. Systems biology approaches using high-throughput technologies may help with mechanistic dissection of host-microbial interactions. Here, we have undertaken a deep phenomics study on four adults receiving sequential FMT for SFCDI, in which we performed a longitudinal, integrative analysis of multiple host factors and intestinal microbiome changes. Stool samples were profiled for changes in gut microbiota and metabolites and blood samples for alterations in targeted epigenomic, metabonomic, glycomic, immune proteomic, immunophenotyping, immune functional assays, and T-cell receptor (TCR) repertoires, respectively. We characterised temporal trajectories in gut microbial and host immunometabolic data sets in three responders and one non-responder to sequential FMT. A total of 562 features were used for analysis, of which 78 features were identified, which differed between the responders and the non-responder. The observed dynamic phenotypic changes may potentially suggest immunosenescent signals in the non-responder and may help to underpin the mechanisms accompanying successful FMT, although our study is limited by a small sample size and significant heterogeneity in patient baseline characteristics. Our multi-omics integrative longitudinal analytical approach extends the knowledge regarding mechanisms of efficacy of FMT and highlights preliminary novel signatures, which should be validated in larger studies.

## 1. Introduction

*Clostridioides difficile* infection (CDI) is the most common cause of diarrhea acquired in acute healthcare settings. Hospital-acquired CDI increases healthcare costs 4 times over matched hospitalization, resulting in an added annual cost of $1.1 billion in North America [1,2]. Most patients with CDI have mild to moderate disease and respond to either oral vancomycin or fidaxomicin; a subset of these patients may develop recurrent infections necessitating a different therapeutic approach [3]. Severe infection is defined by an elevated white blood cell count of over 15,000 cells/mL or serum creatinine level >1.5 mg/dL, while fulminant infection is characterized by hypotension or shock, ileus or toxic megacolon [3]. These patients usually require hospital admission for treatment with oral vancomycin and intravenous metronidazole or surgery if refractory to medical therapy. In CDI, perturbation of the gut microbiome has a clear causation in disease pathogenesis, triggered by antibiotic exposure [4,5]. Restitution of the gut microbiome with fecal microbiota transplant (FMT) is highly effective in the treatment of recurrent CDI. To prevent mild-moderate CDI recurrence, a single FMT is administered once vancomycin has been discontinued, and the success rate is in the range of 80–90% [6]. In patients with antibiotic-refractory severe or fulminant CDI (SFCDI) who are poor surgical candidates, treatment options are limited and sequential FMT by colonoscopy with concomitant vancomycin has been shown to be effective in several small case series and a single randomized trial [7,8,9,10]. However, despite its effectiveness, significant knowledge gaps remain in our understanding of how FMT exerts these beneficial effects, and what molecular features, particularly immunological, may predict treatment outcomes in this unique population of antibiotic refractory SFCDI [11].

Previous data generated by our laboratory and collaborative network has demonstrated that successful FMT for mild-moderate CDI is associated with significantly decreased stool levels of the primary bile acids chenodeoxycholic acid and cholic acid, and significantly increased levels of the secondary bile acids deoxycholic acid and lithocholic acid [12]. In addition to restoration of gut microbiota and bile acid profiles, these findings were accompanied by increased levels of circulating fibroblast growth factor (FGF)-19, consistent with the upregulation of the farnesoid X receptor (FXR)-fibroblast growth factor (FGF) pathway [12]. Interestingly, microbially mediated production of certain secondary bile acids, which predominate in post-FMT stool [13,14,15], has been demonstrated to promote the generation of peripheral regulatory T cells [16], linking these metabolites with colonic immunity. Furthermore, successful FMT for recurrent CDI is also associated with restoration of short chain fatty acids (SCFA) and inhibition of *C. difficile* growth [17,18]. Similar to secondary bile acids, SCFA can regulate colonic regulatory T cell and modulate regulatory B cell immunosuppressive function in mice, which has been directly demonstrated to be a protective mechanism against colitis [19]. Our group has also detected alterations in the circulating host *N*-glycome with successful FMT for recurrent CDI [20].

Beyond gut microbiota-metabolite changes, there is compelling evidence that the immune response to *C. difficile* is a predominant factor determining clinical outcome. A picture is emerging of an exaggerated host inflammatory immune response particularly in the context of severe disease [21]. By contrast, natural antibody responses to *C. difficile* toxins, the major virulence factors associated with disease pathogenesis in shaping clinical outcomes, have been conflicting [5], suggesting that antibody titers may not be an accurate predictor of natural immunity to CDI and recurrence [22]. Nonetheless, the recent success of bezlotoxumab (a human monoclonal antibody that binds and neutralizes *C. difficile* toxin B) and the promise of ongoing vaccine trials suggest that antibody, B cell, and T cell responses to toxin B (TcdB), in particular, contribute to the protection against CDI [22]. Although most adaptive immunity CDI research has focused on components of humoral immunity, we and others have detected TcdA- and TcdB-specific memory B cell responses in CDI patients and a murine model of *C. difficile* disease recurrence [23,24,25].

Recent technological advancements in high throughput next-generation sequencing technologies and omics-based sciences have expanded our fundamental understanding of the complex biology and pathogenesis of CDI. The study of *Clostridioides difficile*, which represents a severe perturbation of the gut microbiota, and the impact of sequential FMT, provides a unique opportunity to directly interrogate microbiome-metabonome-immune interactions directly in humans. Here, we present the observations of a highly multidimensional analysis in a small case series including four adults with antibiotic refractory SFCDI who were treated with sequential FMT during which blood and stool samples were collected prospectively. We analyzed multiple host factors and intestinal microbiome changes longitudinally and related these biological metrics to clinical outcomes in this distinctive population.

## 2. Materials and Methods

### 2.1. Study Cohort, Treatment Regimen and Outcome Definitions

A total of 4 patients with SFCDI were included in this study, of which 3 patients (patients 1–3) were enrolled into an open-label trial of sequential FMT by enemas with fidaxomicin. Patient 4 was treated with sequential FMT via colonoscopy as part of usual clinical care with vancomycin and metronidazole prior to the open-label study.

Three adult patients with SFCDI unresponsive to vancomycin and metronidazole were eligible for this open label study (NCT03760484) at the University of Alberta. The study protocol was approved by the local research ethics board (Pro81229) and Health Canada (Control#220509). Key exclusion criteria were bowel perforation and planned colectomy. Patients were treated with 2 cycles of FMTs and fidaxomicin; each cycle consists of 3 consecutive days of FMT with fecal slurry (day 1 = 720 mL, day 2 = 260 mL and day 3 = 180 mL) delivered by enema, with concomitant oral fidaxomicin 200 mg twice daily up to 10 days. Worsening clinical symptoms and/or elevation in inflammatory markers, including leukocyte count and C-reactive protein, triggered the second treatment cycle. The protocol was intentionally flexible since it is not always predictable how quickly or slowly a patient with SFCDI may respond to proposed intervention. We wanted to suppress *C. difficile* burden as much as possible with fidaxomicin prior to initiating the next FMT cycle, but we also did not want to persist in a fixed fidaxomicin treatment duration in a treatment cycle if a patient was not clinically improving. With resolution of diarrhea and normalization of inflammatory markers, fidaxomicin was discontinued and a final enema of 180 mL was administered. Each patient was then assessed clinically at 1, 2, 4 and 8 weeks after final FMT. Blood and stool samples were collected at screening, at the end of each treatment cycle, and 1 and 2–4 weeks post-final enema and stored at −80·°C until analyses. Treatment success (response) was defined as resolution of diarrhea without the need for anti-CDI therapy 2 weeks after final FMT. Treatment failure (non-response) was defined as recurrence of diarrhea requiring anti-CDI therapy within 2 weeks of final FMT.

We also included a fourth patient with fulminant CDI who was treated with oral vancomycin and intravenous metronidazole and sequential FMT by colonoscopy every 5–7 days until resolution of pseudomembranous colitis [7]. Blood and stool samples were collected at similar intervals to the fidaxomicin patients.

### 2.2. FMT Preparation

A total of 3 universal stool donors registered with the Edmonton FMT program (2 females aged 56 and 37 years and 1 male aged 31 years) provided stool. The donor screening process complied with Health Canada regulations. Each donation of 100 g of stool was manufactured into 360 mL of fecal slurry as previously described [26].

### 2.3. Multiomics Studies

Whole blood was separated into peripheral blood mononuclear cells (PBMCs) and sera. Sera from all four patients was assayed for targeted epigenomic (microRNA; miR), metabonomic (short-chain fatty acids; SCFA), glycomic (total serum *N*-glycans, IgG Fc *N*-glycopeptides), and immune proteomic (cytokines, chemokines, total isotype, antigen-specific and neutralizing antibodies to *C. difficile* toxins) changes. Multi-parameter flow cytometry was used to profile PBMCs and the circulating PBMC TCRα and TCRβ repertoire in the three patients who were treated with the fidaxomicin protocol. Fecal samples were profiled for microbiota and metabolite (SCFA and bile acids; BAs) changes. Figure 1 describes the analytical pipeline.

### 2.4. 16S rRNA Gene Sequencing

DNA extraction methods are described in the Appendix A. Gene-sequencing sample libraries for 16S rRNA gene were generated via Illumina’s 16S Metagenomic Sequencing Library Preparation Protocol, but with modifications. Amplification was performed of the V1-V2 16S rRNA gene regions from the fecal DNA, using primers as previously described [27]. Products from the index PCR reactions were cleaned and normalized via the SequalPrep Normalization Plate Kit (ThermoFisherSceintific, Waltham, MA, USA; library quantification was performed using the NEBNext Library Quant Kit for Illumina (New England Biolabs, Ipswich, MA, USA). Sequencing data were obtained using paired-end 300 bp chemistry on an Illumina MiSeq (Illumina Inc, San Diego, CA, USA), with MiSeq Reagent Kit v.3 (Illumina). Processing of sequencing data was performed via the DADA2 pipeline as previously described [28], using the SILVA bacterial database v.132 (https://www.arb-silva.de/, accessed on 25 May 2021).

### 2.5. Metabolomic Analysis

Both ultra-performance liquid chromatography-mass spectrometry (UPLC-MS; for the profiling and analysis of fecal bile acids) and gas chromatography-mass spectrometry (GC-MS; for detection, identification and quantification of short chain fatty acids in feces and serum) were performed. Further details are within the Appendix A.

### 2.6. Serum N-Glycome Analysis

Serum *N*-glycans were enzymatically released from proteins by PNGase F, fluorescently labelled with 2-aminobenzamide and cleaned up from the excess of reagents by hydrophilic interaction liquid chromatography solid phase extraction (HILIC-SPE), as previously described [29]. Fluorescently labelled and purified *N*-glycans were separated by HILIC on a Waters BEH Glycan chromatography column, 150 × 2.1 mm i.d., 1.7 μm BEH particles, installed on an Acquity ultra-performance liquid chromatography (UPLC) H-class system (Waters, Mississauga, ON, Canada) under the control of Empower 3 build 3471 software (Waters). Obtained chromatograms were separated into 39 chromatographic peaks, and the amount of *N*-glycans in each chromatographic peak was expressed as a percentage of total integrated area. From 39 directly measured glycan peaks, we calculated 12 derived traits which average particular glycosylation traits across different individual glycan structures and are, consequently, more closely related to individual enzymatic activities and underlying genetic polymorphisms. The derived traits used were the proportion of low branching (LB) and high branching (HB) *N*-glycans; the proportion of a-, mono-, di-, tri- and tetra-galactosylated *N-*glycans (G0, G1, G2, G3 and G4, respectively); and a-, mono-, di-, tri- and tetra-sialylated *N*-glycans (S0, S1, S2, S3 and S4, respectively).

### 2.7. IgG Fc N-Glycopeptides Analysis

Sample preparation and analysis of IgG *N*-glycopeptides was done following a previously described protocol with minor changes [30]. Briefly, IgG was isolated from 90 µL of serum samples by affinity chromatography using a CIM 96-well Protein G monolithic plate (BIA Separations, Ajdovscina, Slovenia) and vacuum manifold. IgG *N*-glycopeptides were prepared by trypsin digestion of 25 µg of IgG isolates and purified with reverse-phase solid phase extraction (RP-SPE) using Chromabond C18 beads suspension applied to the wells of an OF1100 96-well polypropylene filter plate (Orochem Technologies Inc., Naperville, IL, USA) and vacuum manifold. Purified tryptic IgG *N*-glycopeptides were separated and measured on a nanoAcquity chromatographic system (Waters) coupled to a Compact Q-TOF mass spectrometer (Bruker, Bremen, Germany) equipped with an Apollo II source and operated under HyStar software version 3.2. After calibration, the first four isotopic peaks of doubly and triply charged signals belonging to the same glycopeptide species were extracted and summed together, resulting in 20 Fc *N*-glycopeptides per IgG subclass. Signals of interest were normalised to the total area of each IgG subclass.

### 2.8. RT-qPCR for miRNAs

We chose to undertake targeted profiling of a 6-plex panel of miRNAs that we had previously determined to be most upregulated in the circulation of patients with recurrent *C. difficile* infection following successful FMT. These were comprised of Let-7b, miR-16, miR-22-3p, miR-23a-3p, miR-4454, and miR-451a [31]. miRNAs were isolated from 200 µL of serum samples using the miRNeasy Serum/Plasma Kit (217184, Qiagen, Hilden, Germany) upon addition of Serum/Plasma Spike-In Control (219610, Qiagen) according to the manufacturer’s instructions. Eluted RNA from serum samples was further purified and concentrated by using Amicon Ultra YM-3 columns (3 kDa MWCO; UFC5003, Millipore, Burlington, MA, USA). Reverse transcription was performed using a miRCURY LNA RT Kit (339340) and quantitative polymerase chain reaction (qPCR) using a miRCURY LNA SYBR Green PCR Kit (339346) and miRCURY LNA primer sets for hsa-let-7b-5p (YP00204750), hsa-miR-16-5p (YP00205702), hsa-miR-22-3p (YP00204606), hsa-miR-23a-3p (YP00204772), hsa-miR-4454 (YP02114119), hsa-miR-451a (YP02119305), cel-miR-39-3p (YP00203952), RNU1A1 (YP00203909), and 5S rRNA (YP00203906, Qiagen) on a CFX384 real-time PCR detection system (Bio-Rad, Hercules, California, CA, USA). The qPCR conditions applied were 95 °C for 10 min, and 40 cycles of 95 °C for 10 s and 60 °C for 1 min, followed by melting curve analysis. qPCR reactions were performed in quadruplicates, and miRNA levels were normalized against cel-miR-39-3p (spike-in), RNU1A1 and 5S rRNA. Normalized miRNA levels are expressed as ‘Relative to 1st time point’, in comparison with time point 0 (D0-Start) for each patient, or ‘Relative to 001_01-22′, in comparison with the non-responder (patient 1) at D0-Start (set as 1).

### 2.9. Multiplex ELISA for Profiling Cytokine and Multi-Isotype Antibody Responses

Patient sera were analyzed for the quantifications of 37 key biomarkers of inflammation from the TNF superfamily proteins, IFN family proteins, Treg cytokines, MMPs, and immunoglobulins IgG1, IgG2, IgG3, IgG4, lgA, lgM using the Bio-Plex Pro Human Inflammation Panel 1 (171AL001M) and Bio-Plex Pro Human Isotyping Panel (171A3100M, Bio-Rad), respectively. Samples were analyzed in a Bio-Plex 200 System using the Bio-Plex manager software according to manufacturer’s instructions. The concentrations were calculated by standard curves developed in parallel and are expressed as pg/mL for the inflammatory biomarkers and mg/mL for the immunoglobulins.

### 2.10. Antigen-Specific Microarray

We have previously established and validated a multiplex protein microarray system to measure antibodies to specific *C. difficile* antigens in human sera [32,33,34]. Briefly and unless otherwise stated, all target antigens and protein homogenates used in this procedure were diluted to 100 µg/mL in sterile print buffer (1× PBS containing 50 mM trehalose, 0.01% Tween 20) prior to application to the antigen microarrays. Corner marker solution was a 1:100 dilution of anti-mouse antibody-IR680 conjugate (Licor BioSciences, Lincoln, NE, USA) in the print buffer. Information on control and test antigens and full microarray experimental procedures is detailed in the Appendix A.

### 2.11. Toxin Neutralization Assay

VERO cells were seeded at 1 × 10^4^ per well in a 96-well plate in 50 µL phenol red-free DMEM supplemented with 4.5 g/L D-glucose, 584 mg/L L-glutamine, 25 mM HEPES and 10% FBS and incubated for 18–20 h at 37 °C in 5% CO_2_. Sterile-filtered patient serum was serially diluted 2-fold (1:4 to 1:32) in serum-free, phenol red-free DMEM and mixed with either an equal volume of Toxin A at 200 ng/mL or Toxin B at 1 ng/mL for 1 h at 37 °C. The serum-toxin mixtures were added to VERO cells to give a total well volume of 100 µL and plates were incubated for 18 h at 37 °C. The final concentration of Toxin A and Toxin B was 50 ng/mL and 0.25 ng/mL, respectively. All combinations, including negative controls, were carried out in triplicate. Toxin neutralization was assessed by counting the number of rounded cells versus non-rounded cells in one randomly chosen field of view in each well. Data is displayed as the percentage of cells protected against toxicity.

### 2.12. Isolation and Freezing of Peripheral Blood Mononuclear Cells

Peripheral blood mononuclear cells (PBMCs) were isolated from peripheral blood by density centrifugation using Ficoll-Paque PLUS (GE Healthcare, Chicago, IL, USA). Isolated PBMCs were frozen by re-suspending cells in freezing medium consisting of 10% DMSO (Sigma Aldrich, St Louis, MO, USA) in heat-inactivated fetal calf serum (Biosera, Marikina, Philippines and stored at −80 °C.

### 2.13. Immunostaining via Flow Cytometry

Frozen PBMCs were thawed at 37 °C and washed in 10 mL of RPMI containing FCS (10%) (Sigma Aldrich). The pelleted cells were re-suspended in PBS (1 × 10^6^ cells/mL), were stained with combinations of antibodies (Appendix A) for 30 min at 4 °C and followed by 2 washes with PBS. For intracellular transcription factor staining for regulatory and follicular helper T cell staining, cells were surface-stained (anti-human CD3, anti-human CD4) and fixed with Foxp3 Fix Perm solution (eBiosciences, San Diego, CA, USA) for 30 min. This was followed by washing the cells, permeabilization with diluted permeabilization buffer (eBiosciences) and staining with antibodies for anti-human foxp3 and anti-human bcl6 for 30 min at 4 °C followed by 2 washes with PBS.

Samples were acquired using a Cyan ADP flow cytometer (Dako, Glostrup, Denmark). Data analysis was done using Summit V 4.3 software. Spectral overlap when using more than one colour was corrected via compensation. Appropriate isotype controls were used for setting gates. The gating strategy used to identify the T cell subset has been shown in Appendix A; the gating strategy for B cell subset distribution has been published [35,36]. The detailed methods for stimulation of PBMCs to induce cytokine production by CD4 T cells and staining for toxin expressing immune cells can be found in the Appendix A.

### 2.14. RNA Isolation, TCR Library Preparation and Sequencing

Total RNA was isolated from PBMCs using the RNAeasy mini kit (Qiagen) following manufacturer instructions. Starting from up to 1500 ng of total RNA, molecular-barcoded TCR cDNA libraries were prepared as previously described [37], with minor modifications for both TCRα and TCRβ chains. See Appendix A for a detailed protocol. Libraries were pooled using 5 ng per library and sequenced on an Illumina NovaSeq6000 SP 2 × 150 bp flow cell. Custom sequencing primers were added to the Illumina primers.

### 2.15. TCR Data Analysis

PCR and sequencing error correction were performed through identification and selection of unique molecular identifiers using the software MiGEC [38], version 1.2.6. Filtered sequences were aligned on a TCR gene reference, clonotypes were identified and grouped, and CDR3 sequence was identified using the software MiXCR [39], version 2.1.1.

Further analysis was performed using R software and packages Vegan [40] for diversity analysis, Mfuzz [41] for temporal trajectory clustering, and ggplot2 for visualization. For temporal trajectory clustering, the most abundant 50 clonotypes of each patient and time point were selected. Clonotypes present in less than 3 time points were excluded from temporal clustering analysis. For clustering of TCRs together with all other experimental measures, K-means clustering was applied as described below.

### 2.16. Statistical Analysis

Time points for all patients were aligned (D0—Start, D1—Cycle 1, D2—Cycle 2, D3—1WeekPost, D4—2/4WeeksPost), and features observed in the one non-responder patient and at least one of the 3 FMT patient responders were then considered. Valid features were normalized, and responders vs. non-responder were compared using statistical *t*-tests between all time points, early time points (D0–D2), and post-FMT time points (D3–D4). Measurements across time points were combined and evaluated using *t*-tests and did not account for any potential correlation present within individuals. This combination of time points was necessitated due to the single individual in the non-responder group. Features were categorized into two groups: “divergent” features showed no significant difference between responders and non-responder at the early time points but demonstrated significant difference at the post-FMT time-points; “convergent” features showed significant difference at early time points and no difference at later time points. The measurements of the same feature in each group were averaged before clustering. To harmonize the heterogeneous profiles for trendline clustering, min-max scaling was performed to map each feature individually to the same range between zero and one. These change patterns were first categorized by K-means clustering and further grouped into 4 major response models according to whether they increased, decreased, increased then recovered or decreased and then recovered following treatment. Pairwise correlations were performed between selected parameters using Spearman’s rank correlation. The features were reordered for visualization using hierarchical clustering [42]. *p*-values ≤ 0.05 were considered statistically significant. *p*-values were not adjusted for multiple testing due to the small sample size and exploratory nature of this work. A heatmap was generated using MetaboAnalyst (version 5.0, www.metaboanalyst.ca, accessed on 13 June 2021) to visualize the immune parameters in the responders vs. non-responder [43].

## 3. Results

### 3.1. Clinical Outcomes

Four patients treated with sequential FMT and concurrent antibiotics for SFCDI were included. Baseline characteristics of included patients are described in Table 1. A total of 3 patients (patients 1–3) followed a similar treatment protocol with the use of fidaxomicin with FMT by enema, and the fourth patient had FMT by colonoscopy and used vancomycin and metronidazole (Table 1). Information pertaining to multidimensional longitudinal datasets, methodologies employed, and delivery routes for FMT, in addition to treatment cycles and sampling time points, are outlined in Figure 1.

A total of 2 of the 3 fidaxomicin protocol patients achieved CDI resolution (responders) with 2 treatment cycles (patients 2 and 3). The failure case (non-responder; patient 1) had transient resolution of diarrhea after 2 treatment cycles, but diarrhea recurred within 2 weeks after the final FMT. Repeat *C. difficile* toxin testing was positive, requiring long-term vancomycin suppression after treating the active CDI episode with vancomycin and metronidazole. Patient 4 achieved CDI resolution after 5 FMT cycles.

### 3.2. Extensive Multi-Analyte Changes Occur with Sequential FMT

681 features were examined (Appendix A). After quality control and data completion filtering, 562 features were used for analysis. 78 of these were significantly different (*p* < 0.05) between responders and non-responder at all time points (Table 2). The trendlines and flow cytometry plots for selected immune cell features are presented in Figure 2. Of particular interest are selected immune features which showed the highest positive fold changes for the FMT responders: % CD8 naïve T cells (Figure 2B), CD8 naïve:memory T cell ratio, % B cells (Figure 2C), % unswitched memory B cells (Figure 2D), % regulatory B cells (Figure 2E), in addition to other features, including microRNA-451a, microRNA-16 and toxin B IgG (*p* < 0.05); see Table 2. Noteworthy features which showed higher fold change*s* for the non-responder included: *Phascolarctobacterium*, unclassified *Enterobacteriaceae*, *Pseudocitrobacter* and *Enterococcus*; *p* < 0.05.

Features demonstrating convergence and divergence are displayed in Appendix A. A total of 114 immune parameters were profiled by multi-color flow cytometry in patients 1–3 (Appendix A). Further details of gating strategies for CD4 T cell subsets (Appendix A) and antibodies used are detailed below. Temporal trends in the normalized % frequency values of other selected immune cytometric parameters for responders (patient 2 and 3 combined) and non-responder (patient 1) are presented in Figure 3. Higher normalized % frequencies of CD4 T cells and naïve CD4 T cells were observed in the FMT responders. In contrast, the non-responder’s peripheral immune system was composed of higher frequencies of total T cells, CD8 T cells, memory CD4 and CD8 T cells, as well as senescent CD4 and CD8 T cell populations. Of interest, circulating senescent T cells (CD28^−ve^CD57^+ve^CD4 T cell and NKG2D^+ve^ CD4 T cell) peaked at the end of cycle 2 and final FMT timepoints, particularly in the non-responder (Figure 3). Peripheral frequencies of anti-inflammatory regulatory T cells peaked at the final FMT time point in the responders (Figure 3). Functional immune assays revealed a higher proportion of anti-inflammatory cytokine IL4 production by CD4 T cells (Th2) and CD8 T cells post-stimulation in the FMT responders, particularly following FMT treatment (Figure 3). Furthermore, anti-inflammatory cytokine IL10 production by CD4 T cells peaked at the end of cycle 1 and was detectable in the circulation of the non-responder until the end of the final FMT, when the patient relapsed, and thereafter fell. In the responders, IL10-producing CD4 T cells peaked in frequency at the end of final FMT before being maintained at a lower but detectable frequency, at least until 2–4 weeks post FMT (Figure 3).

In terms of functional antibody responses, interestingly, only patient 2 of the 4 patients assessed displayed neutralizing anti-*C. difficile* toxin antibodies in their sera (Figure 4), and thus, readouts from this assay were not incorporated into integrative analyses.

Further analysis of toxin-expressing immune cells revealed that the FMT responders were characterized by a decline in peripheral frequency of Toxin A-expressing naïve CD4 T cells (Appendix A) and Toxin A-expressing memory CD4 T cells (Appendix A) at the end of cycle 1 and post-final FMT, respectively. Similarly, a decline in peripheral Toxin A- and B- expressing total B cells (Appendix A) and memory B cells (Appendix A) occurred post-second cycle of FMT; the lowest frequencies were observed post-final FMT cycle and thereafter increased 1 week post-FMT.

Changes in stool microbiota at the class, order, family and genus levels for all patients are described in Appendix A. At the phylum level, responders’ stool microbiota shifted from Proteobacteria predominance at baseline to Bacteroidetes predominance 2–4 weeks after treatment. Conversely, the non-responder demonstrated only a modest decrease in Proteobacteria and no consistent enrichment of Bacteroidetes (Figure 5).

Stool donors’ microbial composition was predominated by Firmicutes and Bacteroidetes. Severe or fulminant *Clostridioides* patients had higher relative abundance of Proteobacteria or Actinobacteria prior to fecal microbiota transplantation (FMT). In treatment responders (patient 2, 3 and 4), the relative abundance of Proteobacteria reduced, but this did not occur in the non-responder (patient 1) after sequential FMT.

### 3.3. Multiomics Longitudinal Patterns Possibly Associated with FMT Response

Analyzed features from FMT responders were longitudinally clustered and classified into 13 temporal behavioral clusters, S0–S12. In total, 4 clusters (S2, S5, S9 and S11) included features which increased, while clusters S10 and S12 decreased over the course of the study intervention (Figure 6A and Appendix A). It was interesting to note that while some clusters were dominated by certain categories of features, others contained mixed feature groups with correlating behaviors. Among these, cluster S10 was particularly noteworthy. This cluster contained features which sharply decreased in abundance from the end of cycle 1, and comprised serum pro-inflammatory proteins; frequency of circulating CD8 memory T cells; frequencies of senescent CD57^+ve^ T cells and CD28^−ve^CD57^+ve^CD8 T cells; IFN^+ve^ and IL17^+ve^ stimulated CD4 and CD8 T cells; IL-4 and IL-10 expression levels in CD8 and CD4 T cells, respectively; disialylated and trigalactosylated glycans; IgG Fc *N*-glycopeptides; fecal 2-hydroxybutyrate; tauro- and glycoconjugated fecal bile acids; as well as bacterial genera comprising *Cutibacterium, Collinsella, Barnesiella, Prevotella_7*, S5-A14a, *Tyzzerella_4*, *Fastidiosipila, Ruminoccocus_1, Phascolarctobacterium, Suterella, Citrobacter*, and unclassified *Mollicutes_RF39*.

The multiomics profile of the non-responder was classified into 12 clusters, F0-F11 (Figure 6B and Appendix A). Of note, features within cluster F1 of the non-responder increased at the end of cycle 2, when the patient transiently improved clinically, before falling to low pre-intervention baseline levels after 1–2 weeks post-FMT when the patient became feverish and tested positive for fecal *C. difficile* toxin. This cluster was comprised of serum butyrate, isobutyrate and fecal 2-hydroxybutyrate; several fecal bile acids; peripheral memory T cells; plasmablasts; IL-4-producing CD8 T cells; and bacterial genera *Faecalibacterium, Fusobacterium* and *Klebsiella*. Other notable clusters (F7 and F10) comprised features which gradually increased after the end of FMT cycle 2. F10 contained serum acetate, serum microRNA-4454, switched memory B cells, tauro- and glyco-conjugated fecal bile acids, *Enterococcus*, *Agathobacter*, *Tyzzerella*, *Dialister*, *Dorea*, *Collinsella*, and members of the *Ruminococcaceae* family.

### 3.4. FMT Impact on T Cell Receptor Repertoire and Multiomics Integration

To investigate the T cell immune response more specifically, longitudinal TCRα and TCRβ repertoire analysis was performed in PBMCS in patients 1–3 (no PBMCs available for patient 4) and two FMT donors. Both FMT donors displayed lower clonality and higher diversity compared to all patients. Both responders (patients 2 and 3) exhibited stable clonality profiles over time. Interestingly, clonality was much higher at screening (pre-FMT) but drastically declined following the first treatment cycle when diarrhea resolved transiently in the only non-responder (patient 1) (Figure 7A). The most abundant TCRs for each patient were clustered based on their temporal behavior using Mfuzz R software. Clusters were identified which appeared to correlate with therapy response, based on the observation that they contained TCRs which sharply increased or decreased after FMT in patients 2 and 3. However, patient 1 (non-responder) did not show clear temporal trajectories in any cluster, but was rather characterized by an intermittent or fluctuating TCR temporal profile (Figure 7B,C and Appendix A).

A dedicated integrative multiomics analysis was performed separately for each patient to include the TCR information, which is unique to each individual. As described in the multiomics paragraph, K-means clustering was performed, and TCRs showing correlative temporal behavior with other increasing-decreasing features were identified (Appendix A).

### 3.5. Temporal Correlation among Features: A Closer Look at T Cell Immunosenescence Signatures, Gut Microbiome and Immunometabolic Features

As an exploratory analysis to identify potentially correlating features which may be associated with treatment outcome, Spearman’s rank correlation coefficients were assessed between selected features of interest for disease pathogenesis and progression and all omics measurements in patients 2 and 3 (FMT responders) (Appendix A) and patient 1 (non-responder) (Appendix A). Of particular interest was the correlation analysis of peripheral senescent T cells (Appendix A). Observations in the non-responder included strong positive associations of peripheral senescent T cells and several host factors, including fecal butyrate; serum hydroxybutyrate; fecal urso-, iso- and hyodeoxycholic acids; serum IgG Fc N-glycopeptides; and microbial taxa, particularly *Pseudomonas* at the genus*,* family and order level*, Coprococcus_1* and *Ruminococcaceae_014*, *Solibacterium*, and *Mollicutes*, which were also positively associated with titers of serum IgA anti-toxin B.

## 4. Discussion

In this longitudinal multiomics study, we observed temporal changes in immune, metabolic and gut bacterial features in a small cohort of patients with antibiotic refractory SFCDI during sequential FMT. We identified 78 features which could help in differentiating responders from the non-responder. Our results hint that non-response may be associated with immunosenescent signals in the non-responder, including: a higher frequency of circulating senescent T cells characterized by loss of CD28 surface antigen and acquisition of CD57; lower B cell and regulatory B cell frequencies; and higher levels of MMP-2, TWEAK, IL-26, sTNF-R1, sTNF-R2, and effector memory CD8 T cells. Furthermore, a higher relative abundance of *Enterococcus,* unclassified *Enterobacteriaceae*, and a lower peripheral TCR repertoire diversity coincided with CDI recurrence, higher levels of fecal primary BAs, and agalactosylated serum IgG *N*-glycopeptides. In support of these immune aging-related observations, enhanced expression of CD57 on CD8^+ve^ T cells has been linked to rejection in renal transplant recipients, highlighting a role of immunosenescence beyond CDI [44]. Aging is accompanied by the decline of CD28 expression on CD4 and CD8 T cells, loss of naïve cells, accumulation of memory T cells, and reduced diversity of the TCR repertoire [45]. The increased secretion of circulating pro-inflammatory molecules, such as MMP-2, sTNF-R1, and IL-26, the master regulator of inflammation in the non-responder, provides further evidence of the senescence-associated secretory phenotype (SASP) [46] and suggests immunosenescence may be a critical driver of treatment outcome in antibiotic refractory CDI patients receiving FMT. Indeed, IL-26 is known to induce the production of SASP proteins, including IL-1β, IL-6, and TNFα, by human monocytes and can trigger NK cell activation, inducing expression of IL-1β, TNFα and type 1 and 2 interferons [47].

The positive correlations observed separately between butyrate, *Pseudomonas* and T cell senescence in the non-responder is indirectly supported by recent studies which demonstrate the ability of sodium butyrate and *Pseudomonas aeruginosa* to induce cellular senescence in human glioblastoma cells and murine lung tissues [48,49], and supports the contention that immunosenescence in the FMT non-responder may be driven by persistent infections; the parallel and increased detection of anti-CMV IgG antibodies in the non-responder provides additional support for this hypothesis. Cytomegalovirus may contribute markedly to immune dysfunction with age [50] and CMV infection has emerged as a pathogenic accelerant of immune cell proliferation underlying immune senescence [51,52]. Human CMV can also cause poor outcomes in solid organ transplant recipients [53,54], and thus, anti-CMV antibody status may be useful in identifying the risk of immunosenescence and FMT failure. Our correlation analysis showing positive associations between senescent T cell frequencies and the *Rikenellaceae* family, which includes the *Alistipes* genus, are in keeping with previous human studies which report that both are over-represented in older adults [55].

Consistent with our findings, bacterially-derived primary bile acids (BAs), such as taurocholic acid, promote *C. difficile* spore germination, while secondary BA inhibit vegetative growth [56] and downregulate toxin activity [57]. Following successful FMT for recurrent mild CDI, we and others have demonstrated that primary BA diminish and are replaced by protective secondary BA [12,58,59]. Secondary BA deficiency in CDI and other forms of colitis “cross-talk” to the host via changes in activation of TGR5- and FXR-mediated signaling pathways [60]. SCFAs stably increase following FMT [17,18,59] and exert an epigenetic effect on the host through the inhibition of histone deacetylases (HDACs); the transcriptional changes arising through this inhibition result in a net anti-inflammatory effect [61]. Valerate is among the SCFAs that have been demonstrated to diminish colitis via HDAC inhibition [61]. Our results also align with our previous study, in which we reported a significantly higher relative abundance of monosialylated and digalactosylated serum *N*-glycans following successful FMT for rCDI [20], supporting a protective role for these serum *N*-glycans. Our observation of a higher relative abundance of circulating IgG4 glycopeptides with agalactosylated and bisected *N*-glycans with core fucose in non-response is in keeping with pathogenic pro-inflammatory and aging states [62].

In contrast, the increase in CD8 naïve T cells in responders supports the notion that these ‘foot soldiers’ of the immune system may be important players in CDI resolution, although the role of the memory T cell response in CDI remains incompletely understood. The detection of more abundant circulating B cells in responders is congruent with a previous report which showed that mucosal IgA-producing B cells are reduced in patients with rCDI [63], and leads us to suggest that FMT may restore B cell frequencies. Nevertheless, only one responder (patient 2) displayed detectable neutralizing antibodies, indicating that other factors, such as the indigenous gut microbiota, may facilitate *C. difficile* clearance, independent of adaptive immunity [64]. However, evidence suggests that, at least in the context of certain viral infections, in vitro toxin neutralization studies may not accurately predict in vivo protection [65,66]. Moreover, it is likely that study participants could have been infected with a *C. difficile* strain other than the historical VPI-10463 strain, from which toxins were derived for in vitro assays described herein. As such, their memory B cell-encoded antibodies may not have been able to neutralize the VPI-10463 toxins. Toxin concentration and the availability of neutralizing antibodies in the gut lumen as well as the rate of toxin clearance may influence disease progression, severity and treatment of CDI [67,68,69]. Interestingly, circulating toxin A- and B-expressing memory B cell frequencies fell at the end of final FMT for the responders before recovering to lower frequencies than those detected prior to FMT. These antigen-specific memory B cells may no longer be required in the context of presumed *C. difficile* clearance and may have transited into the lymphoid organs. Alternatively, these observations may reflect memory B cell apoptosis, which has been reported to contribute to impaired germinal center formation of memory B cells after allogeneic hematopoietic stem cell transplantation [70]. In contrast, peripheral toxin A- and B-expressing memory B cells seemed to peak in frequency two weeks post-FMT for the non-responder. It is possible that a lack of T cell help may have limited memory B cell expansion and differentiation into antibody-secreting plasma cells in this non-responder, or that their toxin A- and B-expressing memory B cells, which increased 2 weeks post-final FMT, only encoded low-affinity antibodies incapable of toxin neutralization. In support of this latter hypothesis, Shah and colleagues recently demonstrated that human *C. difficile* toxin-specific memory B cell repertoires encode poorly neutralizing antibodies, with limited class switching and IgM dominance [71]. These findings indicate that memory B cells may be an important factor in *C. difficile* disease recurrence.

FMT responders also exhibited higher fold changes in frequency of tolerogenic regulatory B cell populations and IL-4 producing CD4 T cells across all time points compared to the non-responder. These findings concur with evidence which demonstrates the protective capacity of immunosuppressive Bregs in the prevention of allograft rejection in renal, liver and lung transplantation recipients and in the development of graft-versus-host disease in stem cell recipients, and with increased frequencies of CD4^+ve^ T cells in the colons of FMT-treated mice, respectively [72,73]. Our findings suggest that, in addition to glatiramer acetate, which is known to stimulate functional Bregs [74], IgG1 Fc *N*-glycopeptides with bisected and non-bisected monogalactosylated and monosiaylated glycans without core fucose may help induce this Breg tolerogenic phenotype, whereas certain taxa such as *Acidaminococcaceae*, *Phascolartobacterium* and *Fastiosipila*, in addition to the fecal bile acids 3-alpha-Hydroxy-7 ketolithocholic acid and chenodeoxycholic acid, may negatively regulate Bregs.

Intriguingly, we observed a higher relative abundance of *Enterobacteriaceae* and *Enterococcoceae* in the non-responder, a finding which may indicate that such pro-inflammatory bacterial taxa, which are more abundant in older adults [75], may play a part in driving biological aging. Consistent with these findings, patients with progeria and progeroid mice display an abundance of Proteobacteria. FMT could reverse this effect, restoring secondary BA biosynthesis and enhanced health and lifespan in progeroid mice [76]. Moreover, the defective germinal center response in aged mice can be rescued by replenishing the gut microbiome of aged mice with that of a younger animal [77].

Investigation of specific temporal clusters in the FMT non-responder also showed that *Faecalibacterium*, *Fusobacterium* and *Klebsiella* increased in relative abundance at the time the non-responder temporarily showed signs of clinical improvement. In keeping with our observation, recent research has shown that specific bacterial genera strongly associate with systemic immune cell dynamics, and that gut microbial taxa, including *Faecalibacteruim* as well as *Ruminococcus* 2 and *Akkermansia*, accelerate immune reconstitution after allogeneic haematopoietic cell transplantation [78]. Moreover, members of *Faecalibacterium*, *Ruminococcus* [79] in one study, and *Akkermansia* [80] in another have been associated with better responses to anti-PD-1 immunotherapy.

The main strength of this study is in the deep longitudinal characterization of four patients that have received sequential FMT for severe or fulminant CDI. However, the small sample size of patients and multiple stool donors in this observational study preclude generalization of the present findings. The single non-responder also has multiple comorbidities, including liver cirrhosis, diabetes, and intestinal bypass for morbid obesity, which may contribute to some of the observed differences from responders. For example, one of the features distinguishing responders from the non-responder was decreased CD4:CD8 ratios, which is known to be lower in liver cirrhosis due to lower CD4 counts [81]. Therefore, definitive conclusions regarding discriminating features of treatment outcomes following FMT cannot be drawn. This unique patient population tends to have multiple comorbidities [7], contributing to the heterogenous population included in our case series. Antibiotic refractory SFCDI fortunately occurs infrequently, contributing to our small sample size. Nonetheless, the results we observed in this study may partially explain why cirrhotic patients may have worse CDI outcomes and may provide multiple starting points to further decipher FMT mechanisms of action. Larger multi-center studies are required to confirm these observations and provide additional mechanistic insight.

Together, these cases provide novel and provocative evidence that extensive host-microbial interactions occur following sequential FMT for SFCDI and suggest a potentially important role for immunosenescence in shaping clinical outcomes. Thus, further studies defining mechanistic and functional outputs of commensals are fundamental to the advancement of next-generation biotherapeutics to treat CDI and other disease states that may be associated with immune aging.

## Figures and Tables

**Figure 1 cells-10-03234-f001:**
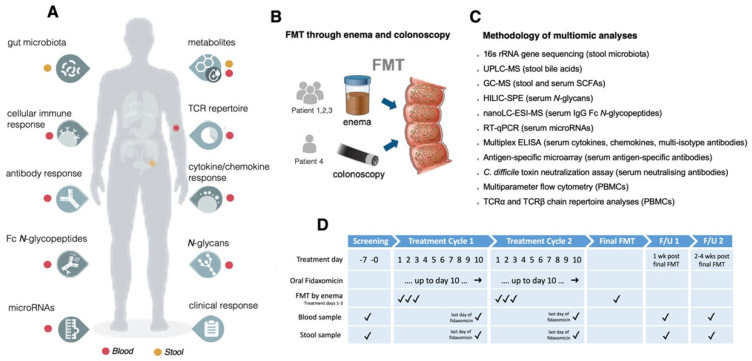
Schematic of pipeline. (**A**) Multidimensional, longitudinal assays performed in patients receiving sequential fecal microbiota transplantation (FMT) either by enema or colonoscopy with severe or fulminant *Clostridioides difficile* infection; (**B**) Delivery route for each patient; (**C**) Methodologies utilized; (**D**) Treatment and sampling strategy with timelines.

**Figure 2 cells-10-03234-f002:**
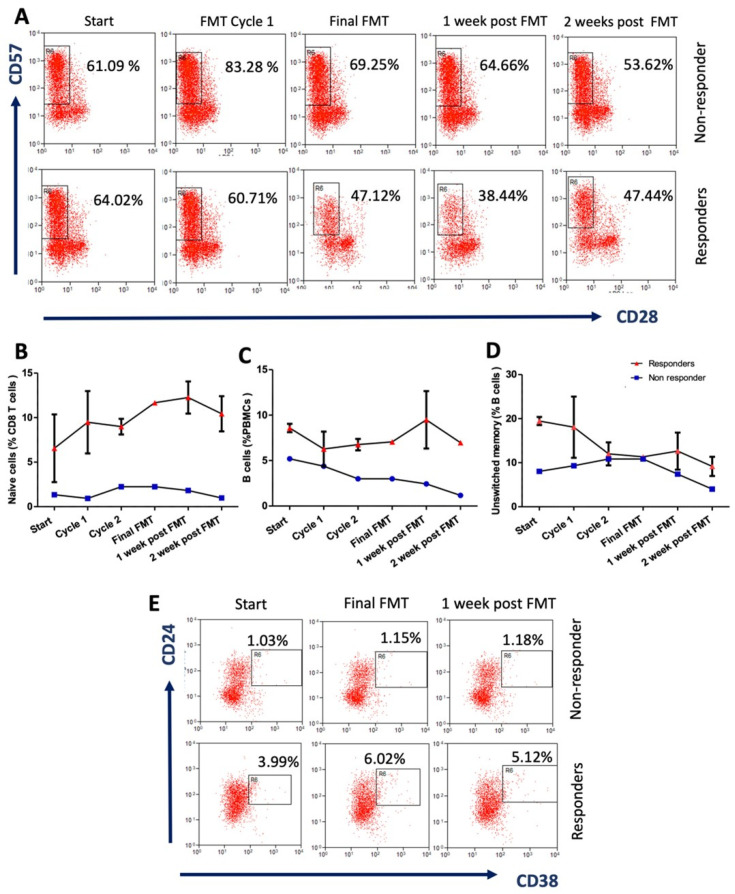
Reversal of immunosenescence features in patients with severe or fulminant *Clostridioides difficile* infection post-sequential FMT. (**A**) Representative flow cytometry plots show the kinetics of peripheral CD28^−ve^ CD57^+ve^ senescent CD8 T cells in FMT responders (*n* = 2) (mean ± S.D data for patient’s 2 and 3) and a non-responder patient (*n* = 1) (patient 1). Percentage of peripheral (**B**) naïve CD8 T cells; (**C**) B cells; and (**D**) Unswitched memory B cells in responders at the start of the trial, post-FMT cycle 1, post-FMT C\cycle 2, post-final FMT cycle, and 1 week and 2 weeks after FMT. (**E**) Representative flow cytometry plots show the kinetics of peripheral CD24^hi^ CD38^hi^ regulatory B cells in FMT responders and non-responder patient at the start of the trial, post final FMT cycle and 1 week after FMT.

**Figure 3 cells-10-03234-f003:**
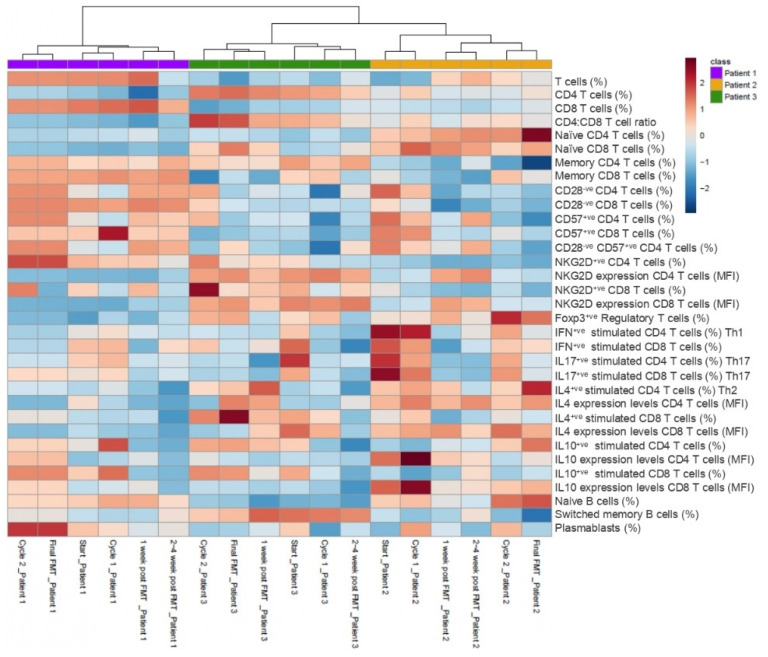
Heat map of normalized frequency values of selected immune subset parameters in the responders (patient 2 and 3 combined) and non-responder patient (patient 1) at different time points. Patients 1, 2 and 3 were clustered using hierarchical clustering (Euclidian distance based). High and low normalized frequency values are indicated in red and blue, respectively. Different immune subset percentages for FMT responders (*n* = 2; patients 2 and 3)) and non-responder (*n* = 1; patient 1) are indicated at the end of the column.

**Figure 4 cells-10-03234-f004:**
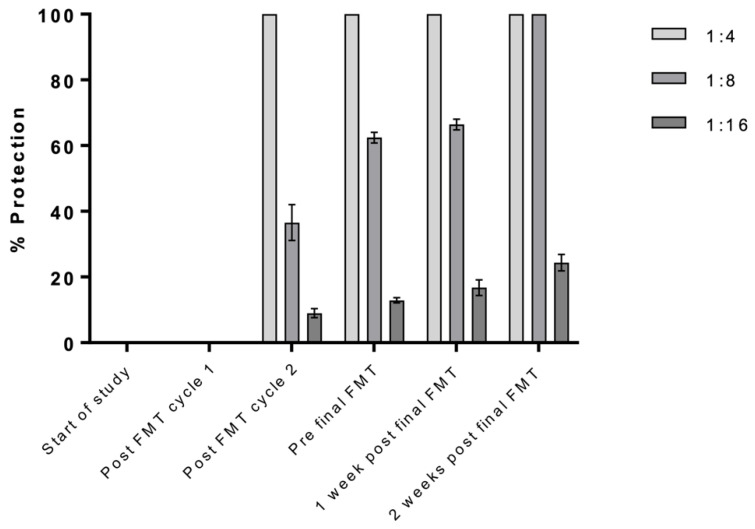
In vitro evaluation of antibody-mediated neutralization of Toxin B (patient 2, FMT responder). Sera were serially diluted and incubated with whole purified toxin B (toxinotype 0, strain VPI 10463, ribotype 087) before addition to VERO cells. Cytotoxicity was assessed by counting the number of rounded and non-rounded healthy cells, expressed as percentage protection. For patient 2, detection of neutralization against Toxin B became apparent post-FMT cycle 2 with 100% protection from Vero cell rounding observed with the most concentrated serum tested (1:4 dilution) compared to 0% at the 2 earlier time points. The degree of protection increased from this point over the course of treatment, with protective efficacy clearly detected even in the lowest dilution tested, 1:16. For 1:8 diluted sera, the mean percentage of healthy, non-rounded, protected cells increased from 36.57% at post-FMT cycle 2 to 62.43% pre-final FMT to 66.42% 1 week post-final FMT to 100% 2 weeks post-final FMT. Unlike this patient, patients 1 and 3 displayed no neutralization against Toxin B, and none of the patients showed neutralization against Toxin A throughout the treatment. Controls for this assay showed 100% rounding for cells incubated with the appropriate toxin alone and 100% healthy non-rounding for cells incubated with the respective serum dilution alone. Sera from patients 1 and 3 showed no neutralization against Toxin B. No neutralization was shown against Toxin A for patients 1, 2 or 3. Data from triplicate values +/− SD.

**Figure 5 cells-10-03234-f005:**
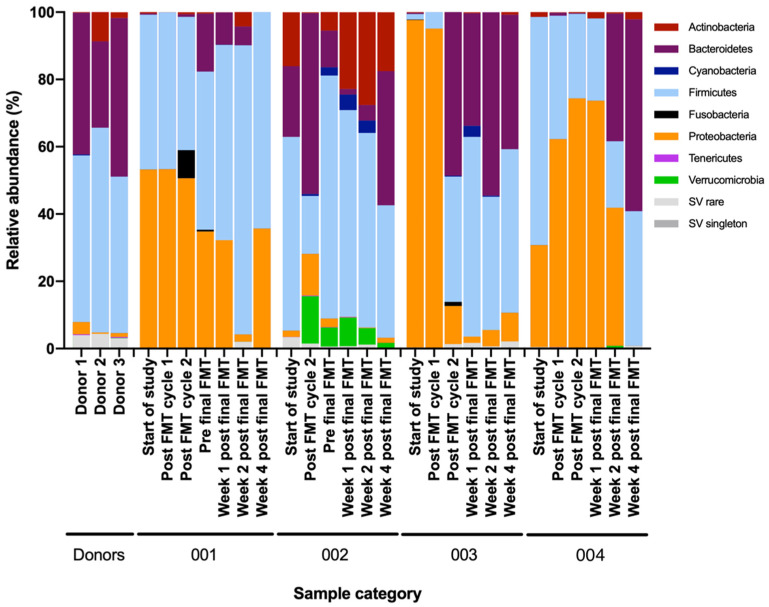
Fecal metataxonomic changes at the phylum level in relation to sequential FMT in severe or fulminant CDI patients. 16S rRNA gene sequencing of DNA extracted from stool samples, presented as relative abundance plots. Participant samples presented as: three stool donors; patient 1(001), earliest to latest timepoint; patient 2 (002), earliest to latest timepoint; patient 3 (003), earliest to latest timepoint; and the fourth patient, earliest to latest timepoint.

**Figure 6 cells-10-03234-f006:**
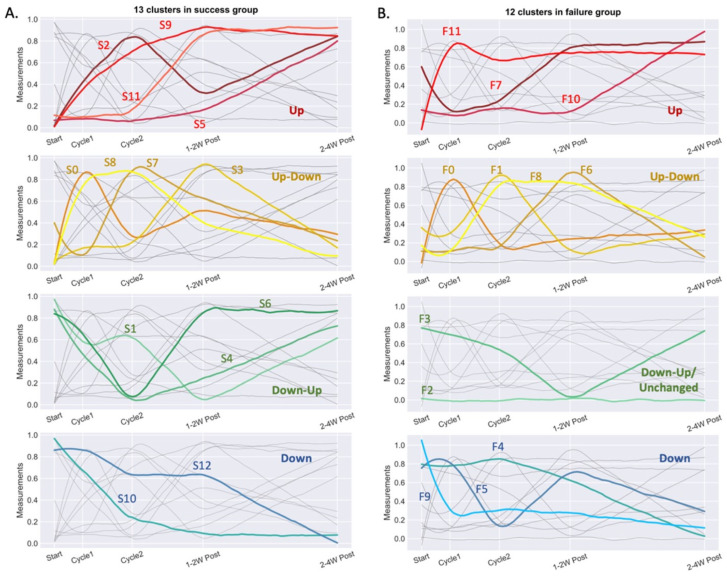
K-means clustering of trendlines of 562 valid measurements. Results shown for FMT responders (*n* = 2) (success group; (**A**)) and FMT non-responder (*n* = 1) (failure case; (**B**)) with each cluster indexed. Clusters are regrouped into four categories as highlighted in each row of the subplots: increased after FMT (up, red), increased after FMT but recovered (up-down, yellow), decreased after FMT but recovered or unchanged (down-up or unchanged, green), and decreased after FMT (down, blue).

**Figure 7 cells-10-03234-f007:**
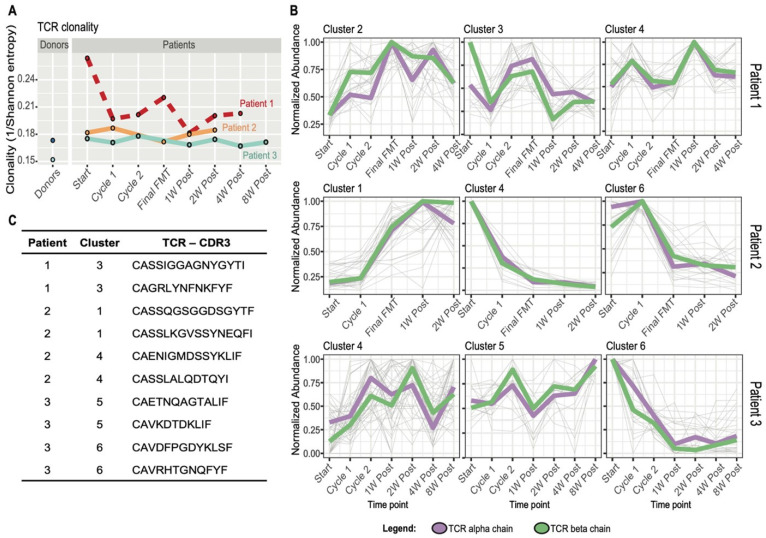
Longitudinal TCR repertoire analysis. (**A**) TCR repertoire clonality calculated as the inverse of the Shannon entropy on sampled peripheral blood mononuclear samples to 1000 TCR sequences. (**B**) Temporal clustering performed with Mfuzz R package for the 50 most abundant TCRs, alpha and beta, for each time point for patient 1 (FMT failure/non-responder), patient 2 (FMT success/responder) and patient 3 (FMT success/responder). The figure shows 3 exemplary clusters out of 6 for each patient. Thin grey lines in the background represent single clonotypes. The median value of the temporal trajectories of TCR alpha (violet) and beta (green) chains for each cluster is displayed. (**C**) Exemplary TCR CDR3 amino acid sequences of the most interesting temporal clusters containing TCRs increasing or decreasing in abundance during and after FMT for patients 2 and 3 and TCRs with high abundance at baseline and during disease recurrence for patient 1.

**Table 1 cells-10-03234-t001:** Participant characteristics and treatment outcomes.

Participant ID	Patient 1	Patient 2	Patient 3	Patient 4
Sex	Male	Female	Female	Male
Age	70	61	85	84
Comorbidities	Chronic pain, NASH cirrhosis (MELD score 9), bariatric Roux-en-Y surgery, chronic obstructive pulmonary disease, depression, atrial fibrillation, hypothyroidism	Congenital blindness in left eye, anxiety	Hypertension, moderate aortic stenosis, oseoarthritis	Hypothyroidism, type 2 diabetes, hypertension, myocardial infarction abdominal aortic aneurysm, benign prostatic hypertrophy, chronic kidney disease, prior laparotomy for diverticulosis and small bowel obstruction
Pertinent Medications	Hydromorphone, Flomax, Furosemide, Breo-Ellipta, Synthroid, Apixaban	Temazepam, Citalopram, Gabapentin	Rosuvastatin, Perindopril, Hydrochlorothiazide	Lipitor, Fenofibrate, Synthroid, Lopressor, Flomax
Treatment outcome	Failure	Success	Success	Success
Number of prior CDI	4	1	None	None
FMT prior to study enrolment	>5	None	None	None
CDI Severity	Fulminant	Fulminant	Severe	Fulminant
Anti-CDI Antibiotics during treatment cycles (Total Days)	Fidaxomicin (16 days)	Fidaxomicin (18 days)	Fidaxomicin (11 days)	Metronidazole IV and Vancomycin PO (25 days)
Number of treatment cycles *	2	2	2	5 FMTs by colonoscopy

* Each treatment cycle consists of 3 consecutive daily FMTs by retention enema plus concurrent fidaxomicin for up to 10 days. Patient 4 received FMT by colonoscopy every 5–7 days with concurrent IV metronidazole and oral (PO) vancomycin, until resolution of pseudomembranous colitis. CDI: *C. difficile* infection; FMT: fecal microbiota transplant.

**Table 2 cells-10-03234-t002:** Features demonstrating statistically significant threshold difference between responders and non-responder (average across all timepoints).

Feature	Category	*p*-Value	Fold Change (log2) (Succ/Fail)	Mean Value (Responders)	
Features with higher mean in FMT responders					
Naïve:memory CD8 T cell ratio	Flow cytometry	0.0007	2.8982	0.1109	0.0149
Naïve CD8 T cells (%)	Flow cytometry	0.0005	2.6972	9.4820	1.4620
miR-451a	Serum microRNA	0.0040	2.3704	2.2922	0.4433
Regulatory B cells; Bregs (%)	Flow cytometry	0.0008	2.1593	4.2080	0.9420
Toxin B IgG *	Antigen-specific antibody panel	0.0471	1.4260	8.7778	3.2667
Total B cell (%)	Flow cytometry	0.0007	1.3481	7.6880	3.0200
miR-16	Serum microRNA	0.0071	1.3251	1.5138	0.6042
CD4:CD8 T cell ratio	Flow cytometry	0.0002	1.2064	1.7049	0.7388
IgM	Isotype panel	0.0395	0.9896	1.1970	0.6028
EMRA CD4 T cells (%)	Flow cytometry	0.0390	0.9057	25.0740	13.3840
CD28 expression levels on CD4 T cells (MFI)	Flow cytometry	0.0022	0.8997	44.6770	23.9460
Unswitched memory B cells (%)	Flow cytometry	0.0106	0.8939	15.3640	8.2680
IL4^+ve^ stimulated CD8 T cells (%)	Flow cytometry	0.0323	0.8806	2.0290	1.1020
Glycodeoxycholic acid	Stool bile acids	0.0221	0.7647	6.4343	3.7870
A027 IgM [‘A’ = surface layer proteins (SLP) of ribotype 027]	Antigen-specific antibody panel	0.0240	0.6851	5.1609	3.2099
Stimulated CD4 T cells IL4 expression levels (MFI)	Flow cytometry	0.0277	0.6786	10.0230	6.2620
IL4^+ve^ stimulated CD4 T cells (%)	Flow cytometry	0.0451	0.6227	2.5160	1.6340
Total CD4 T cells (%)	Flow cytometry	0.0001	0.4822	55.8040	39.9500
CD28 expression levels on CD4 T cells (MFI)	Flow cytometry	0.0001	0.4684	28.6430	20.7020
Total memory B cells (%)	Flow cytometry	0.0241	0.4349	51.7350	38.2700
IgGII1H5N4F1S1: IgG2&3 glycopeptide with digalactosylated and monosialylated glycan with core fucose	IgG glycoprofiling	0.0289	0.3593	7.8840	6.1460
Stimulated CD8 T cells IL4 expression levels (MFI)	Flow cytometry	0.0030	0.3565	2.3670	1.8488
IgGII1H4N4F1: IgG2&3 glycopeptide with monogalactosylated glycan with core fucose	IgG glycoprofiling	0.0059	0.3545	14.2693	11.1605
CD28 expression levels on CD8 T cells (MFI)	Flow cytometry	0.0315	0.3306	23.2890	18.5200
IgGIV1H5N4F1: IgG4 glycopeptide with digalactosylated glycan with core fucose	IgG glycoprofiling	0.0471	0.2910	4.0918	3.3443
Monosialylated glycans	Serum glycan traits	0.0014	0.2556	16.6257	13.9260
NKG2D expression levels on CD4 T cells (MFI)	Flow cytometry	0.0003	0.2181	5.2907	4.5483
IgGI1H4N4F1: IgG1 glycopeptide with monogalactosylated glycan with core fucose	IgG glycoprofiling	0.0140	0.1917	20.8725	18.2761
Candida IgM	Antigen-specific antibody panel	0.0009	0.1841	7.2038	6.3409
Senescent CD4 T cells NKG2D expression levels (MFI)	Flow cytometry	0.0011	0.1781	5.2329	4.6252
Digalactosylated glycans	Serum glycan traits	0.0042	0.1467	54.6643	49.3780
MMP-1: matrix metalloproteinase-1	Inflammation panel	0.0072	0.1443	7.3383	6.6399
Low-branching glycans	Serum glycan traits	0.0155	0.1114	72.3240	66.9480
Features with higher mean in FMT non-responder					
*Acidaminococcaceae*	Family	0.0212	−3.6351	0.1625	2.0193
*Phascolarctobacterium*	Genus	0.0212	−3.6351	0.1625	2.0193
*Enterobacteriaceae_unclassified*	Genus	0.0013	−2.2113	1.0058	4.6577
*Pseudocitrobacter*	Genus	0.0080	−2.2079	0.7009	3.2383
*Enterococcaceae*	Family	0.0035	−1.8467	1.4509	5.2185
*Enterococcus*	Genus	0.0035	−1.8467	1.4509	5.2185
L001 IgA (‘L’ = lysates of ribotype 001)	Antigen-specific antibody panel	0.0000	−1.5667	24.6667	73.0667
3-alpha-hydroxy-7,12-diketocholanic acid	Stool bile acids	0.0189	−1.4326	2.0780	5.6094
CMV IgG	Antigen-specific antibody panel	0.0000	−1.0003	3350.2467	6702.0667
Toxin B IgA*	Antigen-specific antibody panel	0.0194	−0.9586	1.5437	3.0000
A001 IgA [‘A’ = surface layer proteins (SLP) of ribotype 001]	Antigen-specific antibody panel	0.0381	−0.8841	2.1246	3.9212
3 dehydrocholic acid	Stool bile acids	0.0082	−0.8481	4.0741	7.3340
Beta muricholic acid	Stool bile acids	0.0004	−0.8238	4.3596	7.7168
A027 IgG [‘A’ = surface layer proteins (SLP) of ribotype 027]	Antigen-specific antibody panel	0.0499	−0.7879	3.1054	5.3618
CD28^−ve^ T cells (%)	Flow cytometry	0.0000	−0.7757	35.0250	59.9640
sTNF-R1: soluble tumor necrosis factor receptor-1	Inflammation panel	0.0019	−0.7465	3079.3947	5166.5200
IL-26	Inflammation panel	0.0267	−0.7191	1044.0507	1718.7220
Integrin^+ve^ dendritic cells (%)	Flow cytometry	0.0000	−0.7071	2.0863	3.4059
sTNF-R2	Inflammation panel	0.0150	−0.6979	1274.3983	2067.2920
Total CD8 T cells (%)	Flow cytometry	0.0000	−0.6956	33.6670	54.5240
12 dehydrocholic acid	Stool bile acids	0.0135	−0.6126	6.7393	10.3044
Chenodeoxycholic acid	Stool bile acids	0.0000	−0.5788	8.4395	12.6054
CD28^−ve^CD57^+ve^ senescent CD8 T cells (%)	Flow cytometry	0.0008	−0.5387	46.6120	67.7120
Antennary fucosylation	Serum glycan traits	0.0017	−0.5289	9.8970	14.2800
CD28^−ve^ senescent CD8 T cells (%)	Flow cytometry	0.0002	−0.4370	61.7170	83.5540
CD28^−ve^CD57^+ve^ senescent CD4 T cells (%)	Flow cytometry	0.0497	−0.4303	23.3810	31.5060
Tetragalactosylated glycans	Serum glycan traits	0.0017	−0.4232	4.9430	6.6280
Cholic acid-3-sulfate	Stool bile acids	0.0362	−0.4122	5.5642	7.4042
IgGIV1H3N5F1: IgG4 glycopeptide with bisected agalactosylated glycan with core fucose	IgG glycoprofiling	0.0161	−0.3958	8.3615	11.0012
Tetrasialylated glycans	Serum glycan traits	0.0074	−0.3942	4.2567	5.5940
Chenodeoxycholic acid-3-sulfate	Stool bile acids	0.0412	−0.3777	7.2848	9.4649
CD8 effector memory T cells (%)	Flow cytometry	0.0009	−0.3580	58.0380	74.3860
CD57^+ve^ senescent CD8 T cells (%)	Flow cytometry	0.0290	−0.3452	55.0520	69.9360
IgGII1H3N4: IgG2&3 glycopeptide with agalactosylated glycan without core fucose	IgG glycoprofiling	0.0051	−0.3291	1.5296	1.9215
High-branching glycans	Serum glycan traits	0.0071	−0.2972	25.0977	30.8400
IgGII1H4N4: IgG2&3 glycopeptide with monogalactosylated glycan without core fucose	IgG glycoprofiling	0.0276	−0.2911	3.3512	4.1005
Trisialylated glycans	Serum glycan traits	0.0207	−0.2873	16.2837	19.8720
Trigalactosylated glycans	Serum glycan traits	0.0327	−0.2646	20.1547	24.2120
Total T cells (%)	Flow cytometry	0.0077	−0.2526	57.9910	69.0860
IgGII1H4N5S1: IgG2&3 glycopeptide with bisected monogalactosylated and monosialylated glycan without core fucose	IgG glycoprofiling	0.0458	−0.2491	1.8785	2.2325
MMP-2: matrix metalloproteinase-2	Inflammation panel	0.0000	−0.2025	9.3344	10.7411
Candida IgG	Antigen-specific antibody panel	0.0001	−0.1961	8.0645	9.2384
Total CD8 memory T cells (%)	Flow cytometry	0.0015	−0.1777	87.0120	98.4160
TWEAK/TNFSF12: TNF-like weak inducer of apoptosis/tumor necrosis factor superfamily	Inflammation panel	0.0033	−0.1684	5.4639	6.1403
Integrin expression levels on dendritic cells (MFI)	Flow cytometry	0.0235	−0.1316	3.7044	4.0582

MFI—Median fluorescence intensity. * Whole toxins A and B purified from toxinotype 0, strain VPI 10463, Public Health England.

## Data Availability

All data can be found in Appendix A or can be made available upon request.

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
