# Peer review of "A Multi-Factorial Observational Study on Sequential Fecal Microbiota Transplant in Patients with Medically Refractory Clostridioides difficile Infection"

_cells, 2021, doi:10.3390/cells10113234_

Round 1

Reviewer 1 Report

The manuscript entitled “A multi-factorial observational study on sequential fecal microbiota transplant in patients with medically refractory Clostridi oides difficile infection” is interesting and deals the mechanisms of efficacy of FMT against SFCDI. Authors explore the high throughput next-generation sequencing and multi-omics integrative longitudinal analytical approach in current concern.

Also some of the key findings of the study could be included in the “abstract” in manuscript. I would recommend this finding for the next level with my approval.  

Author Response

We have added the following sentence to the abstract. “The observed dynamic phenotypic changes may potentially suggest immunosenescent signals in the non-responder, and may help to underpin the mechanisms accompanying successful FMT, although our study is limited by a small sample size and significant heterogeneity in patient baseline characteristics.”

Reviewer 2 Report

The manuscript describes a multi-factorial investigation on sequential fecal microbiota transplant in patients with medically refractory Clostridioides difficile infection. Three patients were considered as responders, while one was non-responders. The authors have done an excellent job to extensively evaluate several factors. Only minor points that may help improve the manuscript.

  1. The patient no. 4 was a responder, but he had to go through 5 FMT rounds, compared to the ones with enema (2 cycles). Do you think it'd affect your analysis?
  2. Fig. 2 B-D, the baseline for responders and non-responders were different. Any explanation?
  3. Fig. 3, as there were only 2 included responders, it'd be clearer to the readers if the heatmap contains all the data from those, rather than using normalized values. The authors may use the supervised clustering for responders and non-responder.

Author Response

Thank you.

  1. Patient #4 was treated with FMT by colonoscopy before we developed this new study protocol. Each cycle within this study protocol consisted of 3 daily FMT enema. Therefore, patients #1-3 received 2 cycles, or 6 FMT enemas, while patient #4 received 5 FMTs by colonoscopy. Therefore, we do not believe that this would have significantly affected our analysis.
  2. The non-responder (patient #1) had liver cirrhosis and 4 prior episodes of C difficile infection (CDI), while responders (patients 2-3) did not have either cirrhosis or prior CDI. Thus, cirrhosis-associated immune dysfunction may potentially explain the baseline differences in immune cell population between non-responder and responders.
  3. Thank you for your suggestion. We agree that including all the patients will provide better clarity to the readers. We have included both the responders (patient 2 and 3) in the Figure 3 and performed a hierarchical clustering using Euclidian distance on patient 1 (non responder), as well as patients 2 and 3.